# Factors Associated with Treatment Outcomes and Pathological Features in Patients with Osteoradionecrosis: A Retrospective Study

**DOI:** 10.3390/ijerph19116565

**Published:** 2022-05-27

**Authors:** Yoshiaki Tadokoro, Takumi Hasegawa, Daisuke Takeda, Aki Murakami, Nanae Yatagai, Eiji Iwata, Izumi Saito, Junya Kusumoto, Masaya Akashi

**Affiliations:** Department of Oral and Maxillofacial Surgery, Kobe University Graduate School of Medicine, Kobe 650-0017, Japan; takodokoro@gmail.com (Y.T.); dsktkd@med.kobe-u.ac.jp (D.T.); natumikan7731@icloud.com (A.M.); y.pipi.1339@gmail.com (N.Y.); iwataeiji19851126@gmail.com (E.I.); isaito222@gmail.com (I.S.); chivalry_2727@yahoo.co.jp (J.K.); akashim@med.kobe-u.ac.jp (M.A.)

**Keywords:** osteoradionecrosis, radiation dose, orocutaneous fistula, sequestration, extensive resection

## Abstract

A standard treatment for osteoradionecrosis (ORN) has not yet been established because of the diversity. Therefore, identifying the risk factors for a poor prognosis is essential. This study retrospectively investigated the factors associated with the prognosis of ORN in 68 patients. Relevant clinical data of all patients were obtained. Of the patients, 16 who underwent extensive surgery underwent histopathological analysis. The necrotic changes of the anterior and posterior margins in the cortical and cancellous bones were investigated. Multivariate analyses showed statistically significant associations between poor prognosis in patients with ORN and high radiation dose (hazard ratio [HR] 1.15), orocutaneous fistula (HR 2.93), and absence of sequestration (HR 2.49). Histopathological analysis showed a viable anterior margin of the middle portion of the cortical bone for all recovered cases; in contrast, most cases (75%) with a poor prognosis showed necrotic changes. The anterior margin of the cancellous bone was viable and resilient to high irradiation, regardless of the prognosis. These results suggest that patients with orocutaneous fistula should receive early surgical intervention, even if the affected area is limited or asymptomatic. In extensive surgery, a sufficient safety margin of necrotic bone, particularly in the anterior region, is required to improve the prognosis.

## 1. Introduction

Radiation therapy (RT) is vital in head and neck cancers (HNCs) for the preservation of biological functions. Of patients with HNCs, 73.9–84.4% will receive RT with or without chemotherapy once in the course of their disease [1]. Osteoradionecrosis (ORN) of the jaw is one of the most severe chronic complications of RT for patients, and it progresses slowly [2]. It is commonly defined as the exposure of irradiated bone through a wound of the overlying skin or mucosa that persists without healing for at least 3 months [3,4,5]. In some cases, ORN can be visualized radiographically without any bony exposure [6,7]. The incidence of ORN in patients with HNCs is estimated to be 2–22%. Recent publications reported that the incidence rate dropped to 5–8% because of developments in the radiation technique [8,9,10]. The mean duration from completion of RT to the onset of ORN is less than 6 months [3] with a range of 22–47 months [11,12]. Most patients with HNCs develop ORN at 2 or 3 years [6,13]. The symptoms of ORN include asymptomatic bone erosion, localized pain, trismus, severe bone and soft-tissue necrosis, bone exposure, orocutaneous fistula, and pathological fracture [14,15]. To assess progression, radiological investigations are required to detect the presence, severity, and extent of ORN [15]. Computed tomography (CT) is necessary to accurately evaluate the extent and severity of ORN, as well as the associated soft-tissue changes [5,15]. The typical CT findings of ORN are cortical disruption, trabecular disorganization, and fragmentation (sequestration) [16].

Conservative therapies such as saline washing, antibiotics, debridement, hyperbaric oxygen therapy, and pharmacotherapy are commonly used for early ORN. Surgical intervention is considered for advanced cases (intractable pain, osteomyelitis, orocutaneous fistula, and pathological fracture) [5,17,18,19]. In advanced cases, segmental or hemi-mandibulectomy with a vascularized free flap reconstruction is performed as a surgical intervention [20]. Matsuo et al. described that the cure rates of ORN treatment were 44% by only conservative therapy and 63% by combination therapy including surgery [21]. The success rate of ORN treatment remains average, and standard treatment for ORN has not yet been established because of the diversity of symptoms and imaging findings. Therefore, identifying the risk factors for poor prognosis is essential [22]. No study has comprehensively analyzed the influence of pre-treatment findings on prognosis. Previous studies have analyzed the histopathological features of patients with ORN who underwent mandibular resection [23,24,25]; however, their relevance to prognosis has not been reported. In this study, we retrospectively investigated the relationships between various symptoms, imaging findings, and the histopathological features of patients with ORN treated at our department and their prognoses.

## 2. Materials and Methods

### 2.1. Patients

This retrospective cohort validation study included 68 patients with ORN who received and underwent CT between July 2015 and March 2020 at the Department of Oral and Maxillofacial Surgery, Kobe University Graduate School of Medicine. The study adhered to the tenets of the Declaration of Helsinki and was approved by the Institutional Review Board of Kobe University Graduate School of Medicine (No. B200392). All patients consented to treatment after being informed of ORN and its associated risks. CT was performed to diagnose and assess the degree of ORN and to rule out local recurrence and metastasis of HNCs. Patients with non-invasive (in situ) HNCs, incomplete clinical charts, or inadequate follow-up data were excluded.

### 2.2. Definition and Classification of ORN

For this study, ORN was defined as the exposure of irradiated bone that had not healed for more than 3 months, and all other diagnoses were excluded [1,2,3,4,5]. As Lyons et al. described [7], we classified ORN into 4 stages: (Stage 1) length of bone affected (damaged or exposed) is <2.5 cm, asymptomatic; (Stage 2) length of bone affected is >2.5 cm, asymptomatic, including pathological fracture or involvement of inferior dental nerve, or both; (Stage 3) length of bone affected is >2.5 cm, symptomatic, no other features despite medical treatment; and (Stage 4) length of bone affected is 2.5 cm, pathological fracture, involvement of the inferior dental nerve, presence of orocutaneous fistula, or a combination.

### 2.3. Data Collection

From the clinical charts of patients with ORN, the following data were collected: age, sex, stage, onset site and trigger, radiation dose, duration of healing or recurrence, tumor primary site, medical history, clinical symptoms, treatment method, CT image findings, and prognosis. The radiation dose to the site of ORN was investigated using the radiation dose distribution map. The resolution of bone exposure and clinical symptoms for >6 months were considered to indicate recovery (good prognosis). The following CT findings were investigated: the presence of osteolysis, osteosclerosis, periosteal reaction, and sequestration (Figure 1 and Figure 2).

### 2.4. Histopathological Analysis

Of the patients who underwent segmental or hemi-mandibulectomy, 16 patients for whom it was possible to perform histopathological analysis were included. Each specimen was decalcified and fixed in formalin but not frozen. The details of the decalcification method were as follows: Formic acid (98%) (Wako, Osaka, Japan) was diluted to 10% with distilled water. Bone specimens were immersed in 10% formic acid with an ion-exchange resin and treated with an ultrasonic histoprocessor Histra-DC (Jokoh, Tokyo, Japan). Thin sections were obtained from the paraffin blocks and stained with hematoxylin and eosin for light microscopy. Image acquisition of whole bone specimens (×4) was performed using a BZ-X 700 bright-field microscope (Keyence, Osaka, Japan) [24]. We analyzed the necrotic changes in the cortical and cancellous bones, as previously reported [10,23,24]; complete obstruction of the Haversian canals in the cortical bone specimens and the absence of osteocyte nuclei within the lacunae in cancellous bone indicated necrosis. Representative specimens are included in Figure 3. All specimens were independently analyzed by two observers (TH and YT). TH is an oral and maxillofacial surgeon with more than 10 years of experience, and YT is a graduate fellow in our department.

### 2.5. Statistical Analysis

Statistical analyses were performed using SPSS version 22.0 (IBM Corp., Armonk, NY, USA) and Ekuseru-Toukei 2012 software (Social Survey Research Information Co., Ltd., Tokyo, Japan). The association of each variable with ORN was analyzed using the Mann–Whitney U nonparametric test for ordinal variables and Fisher’s exact test or chi-squared test for categorical variables. Statistical significance was set at *p* < 0.05. The remaining variables were introduced into a Cox proportional hazards model. The durations from the onset of ORN treatment to the last date of contact with patients who were lost to follow-up, or the date of recurrence were considered.

## 3. Results

### 3.1. Patients’ Clinical Characteristics and CT Image Findings

Patient clinical characteristics are shown in Table 1. Patients with ORN included 55 men and 13 women, and their mean age was 69.2 ± 11.1 (range, 21–90) years. There were no significant differences in the prognosis of ORN related to age, sex, onset site and trigger, duration of healing or recurrence, primary tumor site, and medical history (hypertension, diabetes, and steroid therapy). There were 16, 3, 21, and 28 patients with ORN stages 1 to 4 according to Lyons’ classification, with recovery rates of 62.5%, 0%, 23.8%, and 60.7%, respectively. Patients with advanced ORN, especially those with stages 2 and 3, had significantly more difficult recovery (*p* = 0.012). The mean radiation dose was 66.4 ± 6.9 (range, 30–81) Gy, and the average observation period was 75.8 ± 40 (range, 1–144) months. The radiation dose for the poor prognosis group was significantly higher than that for the good prognosis group (*p* = 0.003); however, there was no significant difference in the prognosis of ORN related to the onset site (mandibular vs. maxilla, or incisal vs. molar place) or primary HNC (oral vs. other cancers). Orocutaneous fistulas were observed in 15 cases (22.1%). Univariate analysis of the clinical symptoms and prognosis demonstrated that orocutaneous fistula was a significant risk factor for a poor prognosis (*p* = 0.021). Pathological fractures and involvement of the inferior dental nerve were not significant risk factors.

The 68 patients were treated using three approaches: conservative treatment (36 cases, 52.9%), minimal debridement (6 cases, 8.8%), and extensive surgery (26 cases, 38.2%); the cure rates were 33.3%, 33.3%, and 69.7%, respectively. There was a statistically significant association between treatment and prognosis was found (*p* = 0.015). The following were determined based on the CT imaging findings: osteolysis (59 cases, 87.5%), osteosclerosis (45 cases, 71.9%), periosteal reaction (4 cases, 3.1%), and sequestration (27 cases, 53.1%). Of the 36 patients with poor prognosis, 31 (86.8%), 22 (66.2%), 3 (5.9%), and 10 (39.7%) had osteolysis, osteosclerosis, periosteal reaction, and sequestration, respectively. Non-sequestration was significantly associated with poor prognosis (*p* = 0.047).

As shown in Table 2, the associations between the variables and prognosis were analyzed using the Cox model. The data showed that the following factors were associated with a higher risk of a poor prognosis in ORN patients: a high radiation dose (hazard ratio [HR] 1.15, 95% confidence interval [CI], 1.04–1.26; *p* = 0.05), orocutaneous fistula (HR 2.93, 95% CI 1.37–6.25; *p* = 0.05), and the absence of sequestration (HR 2.49, 95% CI 1.11–5.62; *p* = 0.28).

### 3.2. Histopathological Analysis

Sixteen mandible specimens from patients who underwent extensive surgery were included in this study, and the results of bone survival are shown in Table 3. There was no difference in results between analysts. Compared with the cases that recovered, the necrotic rate was higher in patients with poor prognoses for every bone level. The anterior margin of the middle portion of the cortical bone was viable for all recovered cases; in contrast, most cases (75%) with poor prognoses showed necrotic changes. The anterior margin of the inferior border of the cortical bone was necrotic for all cases with poor prognoses and 50% of recovered cases. No necrotic change was found in the anterior margin of the cancellous bone, regardless of the prognosis.

## 4. Discussion

ORN is a devastating side effect of RT, and it is considered a public health problem because of the difficulty in healing. To the best of our knowledge, no prior studies have analyzed the associations between prognosis and various clinical symptoms, CT imaging findings, or pathological findings. In the present study, the risk factors for ORN in patients with a poor prognosis were investigated.

Lyons’ classification was used for the ORN staging, as it is based on clinical and radiological signs [7,26]. The treatment plan described by Lyons et al. was applied according to the ORN stage [7]. Surgical management is considered a radical treatment for patients with ORN: sequestration or marginal resection is considered in cases of localized lesions, and aggressive resection followed by free-flap reconstruction is performed for advanced ORN [5,19,27]. The present study found a statistically significant association between the treatment methods and prognosis. All patients with stage 2 ORN receiving conservative therapy, although only three, were unresponsive to antibiotics and had a poor prognosis. The patients were asymptomatic and did not undergo radical treatment. Conversely, the cure rate of stage 4 patients who underwent extensive resection was 69.7%. With the elimination of diseased soft and hard tissues, patients who undergo extensive surgery tend to have a better prognosis. Extensive jawbone resection with immediate free fibula flap reconstruction should be aggressively applied to patients who have not recovered after medical treatment, as well as those who present with advanced lesions [5,19,27].

Several previous studies have reported a relationship between a high radiation dose and ORN incidence [25,28,29,30]. Chang et al. showed that radiation of ≥70 Gy to the entire jawbone was a risk factor for ORN [30]. Adepitan et al. analyzed the specific ORN site and stated that a radiation dose of >60 Gy was predictive of ORN [25]. However, no studies have analyzed the relationship between the radiation dose and ORN prognosis. Table 1 shows that the mean radiation dose for cases with poor prognoses was higher than that for the cases that recovered. Regions with refractory ORN received 66.4 Gy on average. In this cohort study, multivariate analysis showed that high radiation dose was a significant factor associated with prognosis. On the other hand, in this study, only seven of all 68 patients underwent intensity-modulated RT (IMRT); none of the patients who underwent histopathological analysis underwent IMRT. Therefore, the relationship between radiation technique and prognosis was not analyzed. To reduce the radiation dose to the maxilla and mandible, IMRT is more effective than conventional RT [31]. Gomez et al. [32] demonstrated that the incidence of ORN in their population of patients treated with IMRT was very low, at approximately 1% (two of 168 patients). They concluded that IMRT likely offers an advantage over conventional and three-dimensional conformal techniques in preventing this adverse event. In the future, we should conduct a prospective study including a large sample size of patients undergoing IMRT.

An orocutaneous fistula is a histopathological pathway from the oral cavity to the facial cutaneous surface. This refractory complication results in eating difficulties [19,33,34]. A previous study reported that patients with orocutaneous fistulas responded less to conservative treatment [11,19]. Therefore, surgical intervention is considered effective [33,34,35]. In our study, 15 patients experienced this complication, and only surgical treatment was successful in improving their prognoses.

Multivariate analysis showed that orocutaneous fistula was a significant risk factor for a poor prognosis for the first time. Because of the chronic inflammation and lack of soft tissues, the wound healing ability was significantly reduced and refractory in these cases [36]. Our study strongly suggests that ORN patients with orocutaneous fistulas should undergo early extensive resection, even if the affected area of the bone is limited or asymptomatic.

CT imaging is indispensable for diagnosis and treatment planning for patients with ORN [5,15]. As irradiated bone is in a chronic inflammatory state, osteolysis, osteosclerosis, and sequestration are commonly observed [14,37,38]. Cortical disruption and/or disorganization of the trabeculation characterize osteolysis. Osteolysis is a progressive condition characterized by the destruction of bone tissue, and it is associated with soft tissue invasion and pathological fractures [37,39]. Osteosclerosis is characterized by increased bone density. Miyamoto et al. described the mechanism by which radiation damage affects osteocytes and activates osteoblasts, leading to reactive bone consolidation, particularly in the cancellous bone area [38]. The inhibition and destruction of both osteoblastic and osteoclastic mechanisms also cause sequestration. On CT images, sequestration is seen as osseous fragmentation: devitalized bone is separated from the unaffected viable bone [37,39].

These imaging features were not associated with the severity of ORN [40,41]. However, in the present study, multivariate analysis indicated that non-sequestration was a risk factor for a poor prognosis. As sequestration is an indicator of surgical intervention, the complete elimination of lesions is presumed to be difficult [22]. In this case, extensive jawbone resection, including resection of devitalized bone, is indispensable for improving the prognosis.

In irradiated areas, existing bone cells, marrow stem cells, blood vessels, and extracellular elements are directly damaged, causing reduced cellularity and vascularity of oral hard and soft tissues [2,4,13,14]. Activation of radiation-induced fibroblasts leads to irreversible hypoxia, hypovascular and hypocellular progression and consequent ORN progression [4,42,43]. A previous study described necrotic changes in irradiated cortical and cancellous bones based on microscopic findings. Empty lacunae with complete obstruction of the Haversian canals were observed in the cortical bone. Loss of osteocyte nuclei within the lacuna was observed [24].

We previously found that cortical necrosis is more common than necrosis of cancellous bone [24]. A similar trend was observed in the present study. McGregor et al. reported that circulatory disturbance by radiation was likely to be worst in dense bone, followed by bone marrow and periosteum [44]. Cortical bone is denser and has a lower blood supply than cancellous bone [24,44]. The difference in the histological structure and blood supply between these bones may account for this difference.

In the present study, necrotic changes in the cortical bone were more common at the anterior than at the posterior margin. Since the artery branches and narrows towards the periphery, inadequate blood supply easily occurs in the anterior margin [45]. In addition, smaller vessels diminished over time after irradiation, causing necrosis [46]. This could be the reason for the necrotic change differences between these margins.

We previously reported a possible relationship between ORN recurrence and the presence of residual necrotic bone at the resection margin [24]. Zaghi et al. [23] found that the presence of residual necrotic bone at the resection margins was not associated with ORN recurrence. However, in the present study, all forms of recurrence were recognized at the anterior margin, where almost all layers of bone were necrotic. The cases were inadequate, and the presence of residual necrotic bone could not be extracted as a risk factor; however, we found that cortical bone necrosis at the anterior margin was associated with a poor prognosis. Complete extirpation of necrotic bone is sometimes impossible despite resection of the destroyed bone area with a wide safety margin [24]. Determining the excision margins was also difficult because the boundary between the necrotic and non-necrotic areas was unclear. Considering these factors, the extension of the safe-margin resection in the anterior margin is essential to improve the ORN cure rate. No standard management has been established for patients with postoperative recurrence. In the future, we hope to establish an ORN histopathological score for predicting the prognosis and management method.

This study was limited by its retrospective non-matched design, which meant that other risk factors, such as indices of oral hygiene, could not be examined. Additionally, multivariate analysis was performed to decrease the effect of confounding factors as much as possible, and selection bias could not be completely excluded. A large-scale prospective cohort study is needed to evaluate the predictors in patients with ORN.

## 5. Conclusions

In conclusion, we have successfully demonstrated the associations between various risk factors and the prognosis of patients with ORN. Higher radiation dose, orocutaneous fistula, and non-sequestration were risk factors for poor prognosis. In cases with one or more of these factors, extensive surgery is the treatment of choice. In histopathological analysis, we found that residual necrosis of the anterior margin of the cortical bone is associated with ORN recurrence. Sufficient safety margins, particularly in the anterior region, are required for extensive resection.

## Figures and Tables

**Figure 1 ijerph-19-06565-f001:**
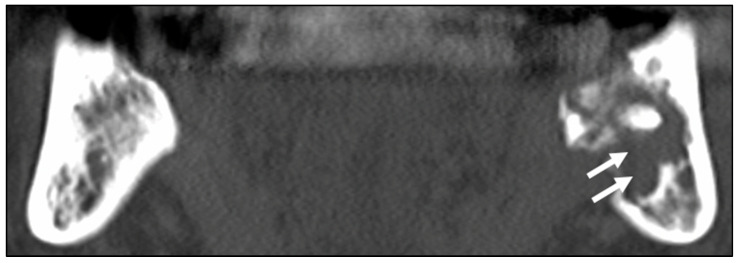
Computed tomography images of a male patient (age 66) who received external radiotherapy (70 Gy) for nasopharyngeal carcinoma and was treated with antibiotics. An extensive osteolytic lesion (white arrows) is seen in the left mandible. Osteosclerosis is observed in the right mandible.

**Figure 2 ijerph-19-06565-f002:**
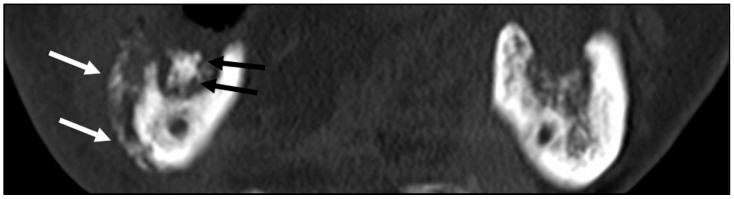
Computed tomography images of a male patient (age 69) who received external radiotherapy (60 Gy) for nasopharyngeal carcinoma and was treated with right segmental mandibulectomy with simultaneous left fibular free flap reconstruction. Periosteal reaction (white arrows) and sequestration (black arrows: devitalized bone) in the right mandible.

**Figure 3 ijerph-19-06565-f003:**
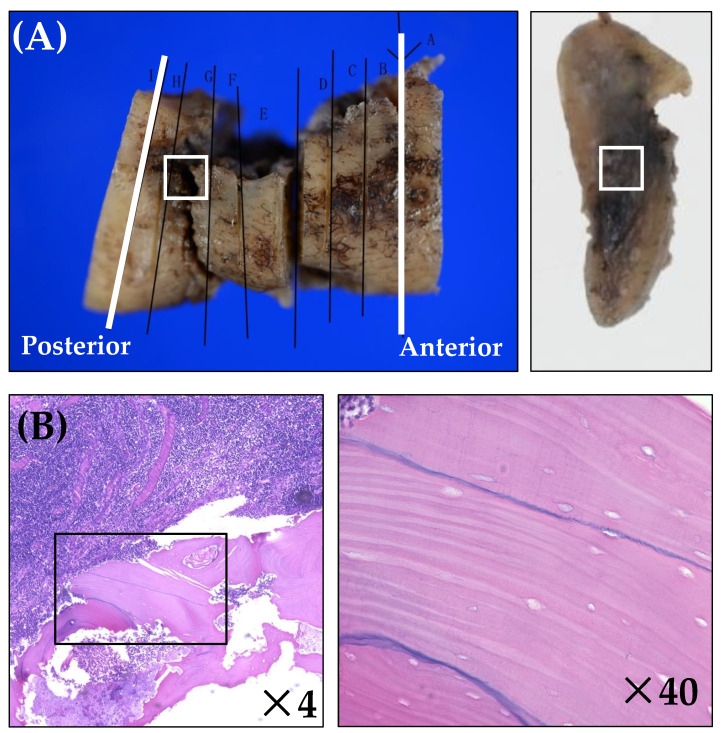
Histopathological images of a male patient (age 83) who received radiotherapy (60 Gy) for oropharyngeal carcinoma and was treated with right segmental mandibulectomy. The specimens were stained with hematoxylin and eosin—original magnification ×4 and ×40. (**A**) A resected bone specimen. The anterior and posterior specimens were prepared separately from the true resection margin to avoid heat artifacts caused by the surgical saw (white lines). The most advanced area of bone destruction was observed in the white box specimen. (**B**) White box specimen in (**A**); the most advanced area of bone destruction (white box). In the enlarged view, osteocyte nuclei within the lacunae are absent. (**C**) Anterior margin. (**D**) Viable cancellous bone. (**E**) The viable cortical bone at the middle level of the mandible. In the enlarged view, there are osteocytes and blood vessels in the Haversian canal. (**F**) Necrotic cortical bone near the inferior border of the mandible. In the enlarged view, the Haversian canal has been completely obstructed. (**G**) Posterior margin. (**H**) Viable cancellous bone. (**I**) The viable cortical bone in the middle portion of the mandible. In the enlarged view, there is a lack of osteocyte nuclei despite the presence of viable blood vessels in the Haversian canal. (**J**) Viable cortical bone near the inferior border of the mandible. In the enlarged view, there are osteocytes and blood vessels in the Haversian canal.

**Table 1 ijerph-19-06565-t001:** Characteristics of patients according to whether osteoradionecrosis led to a poor prognosis.

Variables	Prognosis		*p*-Value
	Good *n* (%)	Poor *n* (%)	
Sex			<0.550 *
Male	27 (84.4)	28 (80.9)	
Female	5 (15.6)	8 (19.1)	
Age			0.431 **
Range (years)	54–84	21–90	
Mean ± SD	69.1 ± 8.0	69.3 ± 13.3	
ORN Stage			<0.013 ***
Stage 1	10 (31.3)	6 (16.7)	
Stage 2	0 (0.0)	3 (8.3)	
Stage 3	5 (15.6)	16 (44.4)	
Stage 4	17 (53.1)	11(30.6)	
Jawbone			<0.886 ***
Maxilla	4 (12.5)	6 (16.7)	
Mandible	25 (78.1)	27 (75.0)	
Maxilla and Mandible	3 (9.4)	3 (8.3)	
Onset Site			0.166 *
Molar region	27 (84.4)	25 (69.4)	
Incisal region	5 (15.6)	11 (30.6)	
Onset Trigger			0.257 ***
Spontaneous onset	20 (62.5)	18 (50.0)	
Tooth extraction	7 (21.9)	8 (22.2)	
Infection	5 (15.6)	6 (16.7)	
Others	0 (0.0)	4 (11.1)	
Primary Tumor Site			0.401 ***
Oral	8 (25.0)	14 (38.9)	
Pharynx	22 (68.8)	19 (52.8)	
Others	2 (6.3)	3 (8.3)	
Radiation Dose			0.003 **
Range (Gy)	30–70	60–81	
Mean ± SD	63.7 ± 9.0	66.4 ± 6.9	
Observation Period			<0.390 **
Range (months)	1–144	0–144	
Mean ± SD	80.8 ± 39.4	71.2 ± 40.6	
Medical History			
Hypertension			<0.550 *
No	24 (75.0)	30 (83.3)	
Yes	8 (25.0)	6 (16.7)	
Diabetes			1.000 *
No	31 (96.9)	35 (97.2)	
Yes	1 (3.1)	1 (2.8)	
Steroid Therapy			
No	29 (90.6)	36 (95.6)	
Yes	3 (9.4)	0 (4.4)	
Clinical Symptoms			1.000 *
Pain			
No	6 (18.8)	6 (16.7)	
Yes	26 (81.3)	30 (83.3)	
Nerve Paralysis			0.182 *
No	21 (65.6)	29 (80.6)	
Yes	11 (34.4)	7 (19.4)	
Pus Discharge			1.000 *
No	13 (40.6)	14 (38.9)	
Yes	19 (59.4)	22 (61.1)	
Pathological Fracture			0.461 *
No	27 (84.4)	33 (91.7)	
Yes	5 (15.6)	3 (8.3)	
Orocutaneous Fistula			0.021 *
No	29 (90.6)	24 (77.9)	
Yes	3 (9.4)	12 (33.3)	
Trismus			0.331 *
No	18 (56.3)	15 (41.7)	
Yes	14 (43.8)	21 (58.3)	
Treatment			0.016 ***
Conservative Therapy	12 (37.5)	24 (66.7)	
Minimal Debridement	2 (6.3)	4 (11.1)	
Resection	18 (56.3)	8 (22.2)	
Osteolysis			1.000 *
No	4 (12.5)	5 (13.9)	
Yes	28 (87.5)	31 (86.1)	
Osteosclerosis			0.443 *
No	9 (28.1)	14 (38.9)	
Yes	23 (71.9)	22 (61.1)	
Periosteal Reaction			0.616 *
No	31 (96.9)	33 (91.7)	
Yes	1 (4.0)	3 (3.1)	
Sequestration			0.047 *
No	15 (46.9)	26 (72.2)	
Yes	17 (53.1)	10 (27.8)	

*: Fisher’s exact test. **: Mann–Whitney U test. ***: Chi-squared test.

**Table 2 ijerph-19-06565-t002:** Results of multivariate logistic regression analysis of the risk factors for ORN.

			95% CI	
Variable	*p*-Value	Hazard Ratio	Lower	Upper
Radiation Dose	0.005	1.148	1.042	1.264
Orocutaneous fistula	0.005	2.929	1.374	6.246
Non-Sequestration	0.028	2.493	1.106	5.624

CI: Confidence interval.

**Table 3 ijerph-19-06565-t003:** Histopathological results of the bone specimens. (V: Viable; N: Necrosis). (**A**) Recovered cases. (**B**) Poor prognosis cases.

Case Number		Anterior Margin		Medial Area (Central Area of Bone Destruction)		Posterior Margin	
	Cortical Bone Inferior Border	Cortical Bone Middle Level	Cancellous Bone	Cancellous Bone	Cortical Bone Inferior Border	Cortical Bone Middle Level	Cancellous Bone
(**A**)							
1	V	V	V	N	V	V	V
2	N	V	V	V	V	V	V
3	V	V	V	N	V	V	V
4	V	V	V	N	N	V	V
5	V	V	V	N	V	V	V
6	N	V	V	N	V	V	V
7	N	V	V	N	V	V	V
8	N	V	V	V	N	N	V
9	V	V	V	V	V	V	V
10	N	V	V	N	V	V	V
11	N	V	V	N	N	N	N
12	V	V	V	N	V	V	N
Necrosis rate (%)	50	0	0	75	25	16.7	16.7
(**B**)							
13	N	N	V	N	N	N	V
14	N	N	V	N	V	V	N
15	N	V	V	N	N	V	N
16	N	N	V	N	N	N	V
Necrosis rate (%)	100	75	0	100	75	50	50

## Data Availability

Not applicable.

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
