# Peer review of "Factors Associated with Treatment Outcomes and Pathological Features in Patients with Osteoradionecrosis: A Retrospective Study"

_ijerph, 2022, doi:10.3390/ijerph19116565_

Round 1
Reviewer 1 Report
The submitted manuscript present a retrospective study about the pathological features due to radionecrosis, after radiotherapy in head and neck cancers. The manuscript is of interest. Following are my comments that in my point of view could improve the manuscript.
Comment1,
Abstract, please also mention in the abstract the total number of patients (68) of the study.
Comment2,
Abstract, lines 21-24. You are referring of the cancerous bone and later in the sentence you mention the anatomical region. Please rearrange the sentence so as the anatomical region as well as orocutaneous fistula is mentioned before ORN
Comment3
Introduction, line 31. Radiation therapy hardly is performed for esthetics. Please delete the word, or put some text explaining it.
Comment4
line 33 throughout the manuscript. It seems you write the reference numbers without brackets. Please check the journal instructions.
Comment5
Materials and Methods patients.
This a major comment of mine. You do, later in the manuscript state the dose received by the patients. However if radiotherapy is considered this is not the only factor affecting side-effects. The authors should check and discuss the radiothepapy technique used, conformal, IGRT, IMRT,... and if possible the treatment isodose curve. Especially for the 16 patients where histopathology data were available. This is important since the area of radionecrosis is associated with the treatment plan
Comment6
line 124. Please state in the manuscript how you choose the sixteen patients for the histopathological analysis (i.e. randomly, were the only available, by using demographic data, etc). If they were the only ones who went extensive surgery please stress it in the manuscript
Comment7
line 131, was BZ-X 700 a fluoresence microscope or something else? Please clarify the type of the instrument.
Comment8
Table 3, There are 'V' and 'N' symbols in table 3. There meaning is confusing please clarify in the table caption their meaning.
thank you
with kind regards
Reviewer 2 Report
The presented manuscript is a retrospective study in which the authors evaluate relationships between various symptoms, imaging findings, and the histopathological features of patients with ORN. The study provides results that document statistically significant association between the treatment methods and prognosis by patients.
The manuscript is suitably elaborated, the introductory passages are suitably introduced into the issue and it is obvious that the authors are experts in the given topic. The methodological procedures are acceptable, the results are described well, the discussion is extensive and satisfactory. I have a few comments on the whole manuscript, which are more or less formal:
- cited sources in the text are not given in brackets, in some places citations are given as a superscript. It is necessary to correct these discrepancies and unify the citations according to the rules of the journal,
- line 13: it written „sixteen patients“, in the following text is discusse about 68 patients (line 60), in the whole text it is necessary to eliminate this discrepancy
- Table 3: due to the extent beyond the side, it is not possible to assess the upper part of the table; it is necessary to adjust the table to the page size
- the manuscript has no conclusion, the conclusion is part of the discussion, it would be appropriate to include the conclusion as a separate chapter and, moreover, to reformulate the conclusion so that it is really a conclusion and not just a repetition of the results
Reviewer 3 Report
Dear Authors, thank you for the interesting paper. Although, it treats with the factors associated with treatment outcomes and pathological features in patients with osteoradionecrosis on humans through a retrospective study it has still some flows and concerns that should be addressed before acceptance for publication. First of all, please read carefully the journal author guidelines and adapt the manuscript. For e.g the author affiliations.
Abstract: please report a sentences or more of state of the art and the necessity of this study at the beginning of the section. Moreover, what kind of surgery did the patients undergo? There are specific sites in which osteoradionecrosis is more commonly seen and it is difficult to understand it on the abstract section. thus, it is messy and not immediately understood by the reader. Please report the references as are requested by the journal author guidelines. the introduction section is too skinny it should be improved with other important results of the worldwide literature. In the abstract is reported that sixteen patients who underwent extensive surgery underwent histopathological analysis. Meanwhile in the materials and methods sections it is reported that 68 patients were included. This sections must be improved too. the details of the Histopathological analysis are not well described. Please improve them. Line 127-129: what do you mean with: several weeks. Could you be more precise on describing the protocol in order to be reproducible by other authors once they try? The sample size is not calculated and reported., Thus how do the authors think that 68 patients are enough from a statistical point of view significance? The images should be inserted into panels for a better comprehension. Here are reported in a messy way. In some images the identifying letters are in capitals in others no. Please correct. Table 3: what means "V" and "N" ? Are they for vital and necrosis? please report.
Reviewer 4 Report
Dear Authors,
Very interesting and important article. To better represent the subject I suggest editorial corrections. There are some spelling mistakes and references should be in a brackets. This mistakes could be easily corrected before acceptance and do not lower the value of presented manuscript.
With regards,
The reviewer
Round 2
Reviewer 1 Report
OK,
Thank you for your response